# RHAMM Is a Multifunctional Protein That Regulates Cancer Progression

**DOI:** 10.3390/ijms221910313

**Published:** 2021-09-24

**Authors:** Britney J. Messam, Cornelia Tolg, James B. McCarthy, Andrew C. Nelson, Eva A. Turley

**Affiliations:** 1Department of Biochemistry, Schulich School of Medicine and Dentistry, Western University, London, ON N6A 5C1, Canada; bmessam@uwo.ca; 2London Regional Cancer Program, London Health Sciences Centre, Victoria Hospital, London, ON N6A 4L6, Canada; cornelia.tolg@gmail.com; 3Masonic Cancer Center, Department of Laboratory Medicine and Pathology, Minneapolis, MN 55455, USA; mccar001@umn.edu (J.B.M.); nels2055@umn.edu (A.C.N.); 4Departments of Oncology, Biochemistry, and Surgery, Schulich School of Medicine, Western University, London, ON N6A 5C1, Canada

**Keywords:** RHAMM, HA, signaling, domains, isoforms, evolution, cancer, multi-functions

## Abstract

The functional complexity of higher organisms is not easily accounted for by the size of their genomes. Rather, complexity appears to be generated by transcriptional, translational, and post-translational mechanisms and tissue organization that produces a context-dependent response of cells to specific stimuli. One property of gene products that likely increases the ability of cells to respond to stimuli with complexity is the multifunctionality of expressed proteins. Receptor for hyaluronan-mediated motility (RHAMM) is an example of a multifunctional protein that controls differential responses of cells in response-to-injury contexts. Here, we trace its evolution into a sensor-transducer of tissue injury signals in higher organisms through the detection of hyaluronan (HA) that accumulates in injured microenvironments. Our goal is to highlight the domain and isoform structures that generate RHAMM’s function complexity and model approaches for targeting its key functions to control cancer progression.

## 1. Introduction

The one gene-one enzyme hypothesis proposed by Beadle and Tatum in 1941 [1], stating that genes produce protein products with a single function, helped shape our understanding of biochemical processes. However, the concept of protein function has since evolved in complexity, with many studies demonstrating that cell signaling can be highly context-dependent, allowing for the ability of single proteins to exert multiple and often opposing functions depending on cell state (e.g., proliferating, senescent, stem-like), environmental signals and protein subcellular compartmentalization [2,3]. However, classifying multifunctional proteins has been challenging. Traditionally, proteins are named for the function they were originally discovered for, but this classification can have an undesirable effect of focusing and thereby restricting perception of function. Receptor for hyaluronan-mediated motility (RHAMM) is one protein whose multifunctional properties are often overlooked, yet it is emerging as an important hub in signaling networks that regulate the interconnected processes of inflammation, interstitial fibrosis, and neoplasia. The multiple functions of RHAMM depend upon its domain and isoform structure [4,5], subcellular localization [6], as well as tissue [7] and cellular context [8,9,10].

RHAMM was originally discovered for its association with the cell surface, its ability to bind to hyaluronan (HA), and its function in promoting avian and mammalian fibroblast motility [11]. This protein was subsequently, and intensively, investigated for its critical role in HA-mediated regulation of inflammation, fibrosis, and disease progression, notably lung disorders and neoplasia [12,13,14,15,16]. Later, RHAMM was shown to also occur in intracellular compartments, and its known intracellular functions now include regulation of interphase and mitotic spindle microtubule dynamics, centrosome functions, and participation in gene transcription [6]. At first glance, this functional diversity may seem unrelated, but the aim of this review is to provide a context in which these multiple functions of RHAMM coordinate a higher-order response to tissue injury.

Proteins such as RHAMM that perform multiple functions are often referred to as “moonlighting or promiscuous proteins” [17,18], but this assumes there is one dominant or mainstream function while others are extra or subsidiary [3,6]. Often, proteins acquire new functions with evolution, which generates novelty and complexity by building upon pre-existing domain structures [19]. Therefore, we use the term multifunctional [2,18] rather than moonlighting in describing the multiple functions of RHAMM. Here, we posit that RHAMM initially functioned as a heparin-binding protein that was co-opted to also bind to HA when this glycosaminoglycan appeared relatively late in evolution.

## 2. Evolution of the Hyaluronome

The hyaluronan genome (“hyaluronome”), which coordinates the metabolism and functions of HA, is complex, comprising synthases, hyaluronidases, and hyaluronan receptors/binding proteins [12,14]. HA is produced by three integral plasma membrane hyaluronan synthases, HAS1-3, that are moderately tissue-specific in their expression and produce HA polymers of different sizes [20]. Knockout and knockdown strategies have revealed both overlapping and distinct functions of these enzymes. At least seven hyaluronidases (HYAL1-5, TMEM2, and KIAA1199/CEMIP/HYBID) have been described that can depolymerize HA, each differing in the sizes of the HA polymers that they produce [21,22,23]. HA receptors that sense the native and degraded HA polymers and activate intracellular signaling include CD44, RHAMM/HMMR, LYVE1, STAB2, TLR4, and LAYN [24,25,26,27,28,29,30,31]. The extracellular organization of HA, which is also critical to its functions, is achieved by multiple proteins, including aggrecan, link protein, and versican [32,33].

HA is a structurally simple linear polymer composed of repeating units of N-acetyl glucosamine and b-glucuronic acid. Like its sulfated glycosaminoglycan (GAG) cousins, HA has key roles in inflammation, but it is strikingly different from its sulfated GAG cousins in its high viscoelastic properties, its production at the cell membrane, and its polymer size-dependent functions in inflammation [12,14,34]. The viscoelastic properties of HA are considered to have been essential for the development and function of an endoskeleton [35,36,37], while the production of HA at the cell surface coupled with its polymer size-dependent functions was utilized as a rapid and sophisticated danger-sensing mechanism [12,38,39]. Both of these properties were likely critical for the emergence of vertebrates when HA first consistently appeared in evolution [40,41,42] (Figure 1).

Notably, invertebrates and lower organisms do not produce HA (Figure 1) [43], although, they synthesize sulfated GAGs (Figure 1) [44,45,46]. In the early chordate phyla, lancelets, but not tunicates, express a *has2* gene with 57% identity to human *HAS2* (Figure 1). HAS genes are only consistently found in vertebrates, predicting that the capability for producing HA was an important change contributing to vertebrate evolution. HA synthases likely evolved from an ancestral glycosyltransferase that also gave rise to the enzymes that produce related polysaccharides such as chondroitin sulfate, chitin, and even cellulose [40,47]. This is suggested by the following common features: the glycosyltransferases that produce HA, chitin, and cellulose B-1,4 glycan are all integral membrane proteins, have invariant amino acid sequences in their catalytic domains, and can under specific conditions synthesize several polysaccharide products [42,47,48]. For example, mammalian HAS1 can produce chitin, a poly-glucosamine polymer, if supplied with only activated glucosamine (UDP-GlcNAc) [40]. Additionally, a Xenopus glycosyltransferase, DG42, can produce both HA or chitin polymers, where the chitin polymers are used as primers for HA oligosaccharide (O-HA) synthesis [47].

Interestingly, even though they do not produce HA, many invertebrate species express genes orthologous to vertebrate HA receptors. For example, despite the inability to produce HA, tunicates express proteins with a link module (Xlink) that is homologous to the vertebrate CD44 link module [49,50]. Rather than binding HA, Xlink domains bind to heparin, which is a sulfated glycosaminoglycan synthesized by all invertebrates and lower species analyzed to date [45,51]. Xlink domains have functions in cell migration and apoptosis [49]. Thus, the CD44 link module is predicted to have evolved from heparin-binding ancestor proteins that were recruited to bind to HA as a result of domain shuffling [50]. Like HA, the secondary structure of RHAMM is largely helical (Figure 2A). RHAMM orthologues, which are also helical proteins, are found in invertebrate species, but these have low sequence identity with human *RHAMM* (varying between 18 and 30%, shown in ants (*A. heyeri*), Figure 3 and Figure 4). The conservation of RHAMM protein is predicted (predictprotein.org (accessed: 18 September 2021)) [52] to exist primarily in the carboxy-terminal region that contains clusters of basic amino acids (Figure 3A) and that binds to HA in vertebrates (yellow-boxed sequence, Figure 3B). This region of mammalian RHAMM has also been demonstrated to bind to heparin [53] and ERK1 [54] and is highly conserved (>80%) in all vertebrates. In addition to these sequences, vertebrate and invertebrate RHAMM sequences both contain regions of coiled coils, which can perform structural [55] and signaling functions [56], and di-leucine repeat motifs, which are involved in receptor endocytosis [57]. Collectively, these similarities suggest that ancient functions of RHAMM were lectin-like (heparin-binding) and associated with signaling functions that were later adapted to HA signaling in vertebrates. The heparin-binding function of RHAMM was likely adapted to bind to HA during vertebrate evolution. Other known RHAMM domains, assessed by sequence homology with human RHAMM, are not present in invertebrates and only appear later in evolution, with the emergence of vertebrates (Figure 3 and Figure 4). These include N-terminal sequences that mediate binding to microtubules, and the carboxyl-terminal leucine zipper, which binds to TPX2 [58], and likely transcription factors such as E2F1 [59] and Bach1 [60]. The binding repertoire expansion of RHAMM during vertebrate evolution predicts that the multifunctionality of RHAMM has increased over time.

## 3. RHAMM Isoform Structure and Function

The *RHAMM* gene contains 18 exons that can be expressed both as a full-length protein and, like CD44, as multiple isoforms generated by alternative splicing, alternate start codon usage, and/or post-translational protein processing (Figure 5A,B). In general, RHAMM is poorly expressed in normal tissues [61], but expression of the full-length protein and isoforms are dramatically and transiently increased during tissue response to injury [14,24,62], and these are constitutively increased with diseases such as breast cancer, multiple myeloma, and bronchopulmonary dysplasia [24,62,63]. Here, we consider both human and mouse RHAMM processing, since these two species are most frequently used to parse the function of this protein. The extent and type of processing for generating RHAMM isoforms differ in humans and mice (Figure 5A,B).

Human RHAMM (Figure 5A) isoforms are generated by alternative splicing of mRNA (uniprot.org (accessed: 18 September 2021)) [64]. The major alternatively spliced forms of RHAMM are A–D (also referred to as 1–4) (Figure 5A), but alternative splicing of exons 2–4 [65], 4, 5, and 13 have also been reported [63,66]. Alternatively spliced RHAMM isoforms are frequently expressed with human disease, particularly in cancers, including multiple myeloma [66,67] breast [68,69], pancreatic [4], and colon cancers [70]. Notably, both an increase in canonical or full-length RHAMM relative to normal tissue and the isoform balance are deregulated in multiple myeloma [67] and pancreatic [4] and colon tumors [70]. Human cancer cells (e.g., PC3MLN4 prostate cancer cells) express full-length and isoform B, although protein products are not easily resolved from one another using standard SDS-PAGE and Western blots (Figure 5). The functions of these human RHAMM isoforms have not been extensively investigated. Isoform B is commonly expressed in cancers and promotes liver metastasis in a mouse model of islet tumorigenesis as well as in tail vein assays [71]. Mechanistically, the pro-metastasis effect of isoform B is associated with sustained activation of EGFR.

Mouse RHAMM (Figure 5B) is similar to the human counterpart but encodes a 20 aa sequence that is repeated five times in the full-length protein and contains an additional sequence unique to mouse RHAMM. The protein/polysaccharide binding domains located in the N- and C-termini are conserved. Mouse RHAMM protein isoforms differ from human comparators by varying more widely in their molecular weight. Mouse isoforms are referred to as full-length (794 aa) and isoforms X1–4 (Figure 5B). Although X1 and X2 isoforms are generated by alternative mRNA splicing, it is likely that X3 results from alternative start codon usage while X4 derives from a combination of these two mechanisms. Western blot assays using antibodies specific to the N- or C-terminus confirm this pattern of isoform expression. N-terminus-specific antibodies reveal only a full-length protein, while C terminus-specific antibodies detect proteins with molecular weights corresponding to isoforms X3 (72 kDa) and 4 (70 kDa) that are expressed in Ras-transformed mouse fibroblasts in vitro [72]. Notably, X3 and X4 correspond to the molecular weights of RHAMM proteins detected on macrophage cell surfaces by surface biotinylation following tissue injury [73]. The additional smaller protein bands detected by Western blot assays may be post-translationally processed forms.

## 4. RHAMM Multi-Functions and Domains

The complex domain structure of RHAMM contributes to both the multifunctionality of this protein and its subcellular distribution. RHAMM binds to, indirectly associates with, and/or modifies the functions of both polysaccharides and proteins. The binding domains responsible for these functions and their partner interactions have been partially identified. The following have been shown to directly bind to RHAMM: HA [74], heparin [53], TPX2 [6], calmodulin [68], and ERK1 [54] (Figure 2B). RHAMM also indirectly associates with or functions through growth factor receptors such as EGFR [4,8,75,76] and PDGFR [77], HA receptors such as CD44 [10,78], and intracellular signaling proteins such as CTNNB1 [79], BRCA1/AURKA [80], ERK2, MEK1 [54], and E2F1 [59]. These interactions firmly establish both extracellular and intracellular RHAMM as a multifunctional signaling entity.

The first binding interaction described for RHAMM was its association with HA [11] and heparin [53]. Subsequent studies showed that LMW-HA and O-HA signaling during excisional wound repair requires RHAMM expression [81]. Truncation and site mutation experiments localized the HA and heparin-binding domain to an alpha-helical region in the carboxy-terminus that contains two clusters of basic amino acids (klk and klr) separated by a leucine zipper (Figure 2B, yellow) [74]. Substitution of specific basic amino acids in this region ablates HA binding, implicating an electrostatic surface as a driving force for RHAMM/HA interactions [82,83]. Indeed, peptide mimicry analyses predict that the association between HA and the carboxyl-terminal domain of RHAMM is largely ionic rather than structural [83,84]. The presence of a leucine zipper that joins the two basic, alpha-helical regions predict an ability for RHAMM to dimerize. The RHAMM/HA-heparin-binding region, therefore, more closely resembles the GAG binding regions of proteins [85], such as GRO chemokines [86], than the CD44 link module. Like RHAMM, GRO chemokines contain helical surfaces of basic amino acids that are primordial for GAG binding and include residues involved in dimerization contacts [86]. Oligomerization of these proteins can alter the exposure of basic amino acid motifs required for binding to glycosaminoglycans (e.g., BBXB and BBXXB motifs) [87] and/or create a glycosaminoglycan-binding groove [82]. Thus, a capability to dimerize can profoundly affect the glycosaminoglycan binding properties of proteins. Conversely, the binding of glycosaminoglycans to proteins, such as SDF-1, promotes dimerization [88] and heterodimerization [89], which are vital to the bioactivity of certain chemokines. These properties suggest that the leucine zipper (Figure 2B) of RHAMM may similarly affect its association with HA. In terms of cellular functions, RHAMM/HA interactions have been shown to regulate cell motility [11,12], myofibroblast activation into a pro-fibrosis state [90,91,92], macrophage chemotaxis and pro-inflammatory activation [92,93], adipogenesis [90,91,94], fibroblast neoplastic transformation [5], and cancer cell proliferation and survival [95,96]. Importantly, the ability of RHAMM to bind to HA is also required for maintenance of sarcoma transformation in experimental mouse models [5]. These functional consequences have largely been assumed to result from extracellular RHAMM/HA interactions since HA is an extracellular matrix component that is released by plasma membrane synthases, and RHAMM display at the cell surface has been documented by flow cytometry [11,73,97], surface biotinylation [73], and detection of GFP tags in non-permeabilized cells (Figure 6). However, the mechanisms responsible for RHAMM release and the domains required for its secretion are currently a matter of debate since RHAMM lacks a leader sequence for export through the Golgi/ER, a transmembrane domain, or post-translational modifications that permit a direct association with the plasma membrane (e.g., GPI-linkage). Plausible mechanisms to account for the display of RHAMM at the cell surface include unconventional export [98], release from dying cells [99], and/or export via exosomes [100] (Figure 7). Evidence to date suggests that once exported, RHAMM associates with the cell surface through interactions with integral cell surface proteins and thus functions as a co-receptor.

Although HA is largely studied for its extracellular functions, it is now acknowledged to also occur in intracellular compartments including the cytoplasm and nucleus [101], where it co-associates with intracellular RHAMM in the nucleus [97] and on mitotic spindles [102]. The phenotypic consequences of intracellular HA/RHAMM interactions are presently unclear [26], but one likely function in the nucleus is regulation of gene expression, particularly of genes linked to inflammatory pathologies [101].

Several groups have shown that RHAMM also binds to a variety of intracellular proteins. It binds to tubulin (TUBA1A, TUB2B, and TUBG2) [54,103] via two separate domains, which result in an association with both interphase and mitotic spindle microtubules (Figure 2B). The first domain occurs in exon 4 of the N-terminus, while the second requires a leucine zipper embedded in the basic region of the carboxyl-terminus [54,103]. The leucine zipper interacts dynamically with microtubules, while the N-terminal microtubule interaction is more stable (e.g., deletion of exon 4 strongly reduces the association of RHAMM with tubulin) (Figure 5C). Loss of the N-terminal interphase microtubule-binding domain in both mouse and human RHAMM (e.g., mouse isoform X3 and human RHAMM B) is transforming and promotes tumor progression to metastasis, while the full-length protein with an intact microtubule-binding ability does not [5,71,77]. One consequence of the loss of this binding domain is a marked change in the subcellular distribution of RHAMM. Thus, loss of either exon 4 [68] or truncation of N-terminal residues 1–184 shifts RHAMM distribution from interphase microtubules to more diffuse staining and, notably, a presence in the cell nucleus (Figure 5C). Loss of this domain is common in cancer and has also been reported in injured tissues [10,73,104]. Whether the subcellular shift resulting from loss of interphase microtubule binding is mechanistically connected to these processes is not clear. Thus, for both humans and mice, deletion of the N-terminal interphase microtubule-binding domain markedly changes the subcellular distribution of RHAMM and enhances neoplastic initiation and/or metastasis.

The RHAMM carboxy terminus contains a sequence that mediates an association with TPX2 and TUBG2 of the mitotic spindle [58] and with both ERK1,2/MEK1 and TUBA1A/TUB2B. The RHAMM/TPX2/TUBG2 complexes provide cues for the spatial association of AURKA with the spindle that is required for centrosome maturation, entry into mitosis, formation, and function of the bipolar spindle, and cytokinesis [58,105]. The association of RHAMM with mitotic spindles also affects RanGTP-mediated microtubule nucleation [106], spindle orientation [107], normal spindle assembly [54,108], and spindle integrity [54,58]. However, the consequences of these RHAMM/TUBG2 interactions are likely to be context-dependent. For example, less than 10% of mitotic spindles are aberrant in *Rhamm*−/− fibroblasts in vitro [54] and, although the carboxy-terminal mitotic binding region of RHAMM alters spindle orientation in HeLa cells in vitro [109] and granulosa cells in vivo [110], its loss does not affect spindle orientation of neuro-progenitor cells [107]. RHAMM interactions with TPX2 and other proteins also play a role in the localization of the microtubule organizing center (MTOC) and centrosomes. For example, RHAMM/TPX2 complexes are required for centrosome separation in prophase of mitosis [111], while RHAMM/SPTA1 complexes are necessary for the rear polarization of the MTOC, mediating directed smooth muscle cell migration [108]. The RHAMM/ERK1,2/MEK/TUBA1A/TUB2B complexes modulate interphase microtubule dynamics, which requires ERK1,2 activity [54]. RHAMM binds directly to ERK1 via non-canonical docking sequences [54]. Consistent with a binding specificity for this MAP kinase, RHAMM-loss shares a subset of ERK1 knockout phenotypes: neither result in embryonic lethality (although ERK2 knockout does) [112,113], and genomic loss of either gene promotes adipogenesis [91,94,113]. RHAMM binds directly to ERK1 via non-canonical docking sequences [54]. Consistent with a binding specificity for this MAP kinase, RHAMM-loss shares a subset of ERK1 knockout phenotypes: neither result in embryonic lethality (although ERK2 knockout does) [112,113], and genomic loss of either gene promotes adipogenesis [91,94,113]. Intriguingly, the RHAMM human sequence contains eight di-leucine motifs that are often present in the cytoplasmic domains of receptors and are required for protein internalization and trafficking [114,115] and which can play key roles in the spatial and temporal regulation of signaling pathways [57]. The presence of these motifs raises the possibility that cytoplasmic RHAMM may function as an adaptor protein to promote the internalization of receptors. This possibility is intriguing because RHAMM forms complexes with extracellular matrix receptors such as CD44 [10,78,96,116] and growth factor receptors such as EGFR and PDGFR [8,77]. RHAMM also associates with F-actin [117], CTNNB1 [79], and the transcription factors E2F1 [59] and Bach1 [60], although the domains required for these interactions have, to our knowledge, not been reported.

In summary, the domain structure of RHAMM, which is both novel and complex relative to other HA-binding proteins, reveals an impressive repertoire for multiple protein and polysaccharide partners that is estimated to include over 99 binding partners. The functional consequence of these domains is determined by isoform structure.

## 5. RHAMM Multi-Functions in Cancer

It is intriguing to note that RHAMM can, like CD44 [118], have opposing functions that impact oncogenicity. For example, *Rhamm*-loss reduces fibroblast motility but conversely stimulates keratinocyte migration during excisional wound repair in vivo [8]. This wound and cell context-dependent effect appears also to be manifested in diseases such as cancer. For example, either embryonic deletion of *Rhamm* [7] or isoform (RHAMM B) overexpression [4] similarly promote pancreatic cancer in mouse models of susceptibility and xenograft models. Correspondingly, RHAMM knockdown in breast cancer cell lines can either promote or inhibit motility depending upon the molecular subtype [9]. These in vitro results suggest that RHAMM expression status has intrinsic and tumor cell-specific effects depending upon the cell type. However, in the case of pancreatic cancer, the dominance of a pro- or anti-tumor effect of RHAMM may also be influenced by *Rhamm*-loss in the tumor microenvironment [7], in addition to its intrinsic consequences to tumor cell functions, which will affect cancer-associated fibroblasts and immune cell tumorigenic functions.

Most analyses linking RHAMM expression to a prognostic outcome are based upon macro-level analyses of total tumor RHAMM mRNA or protein expression. These show that elevated RHAMM expression is characteristic of most cancers and prognostic of a poor outcome in many. For example, elevated RHAMM expression is an indicator of poor prognosis in breast cancer [119,120,121], multiple myeloma [122], oral squamous cell carcinoma [123], ovarian cancer [124], lung cancer [125], prostate cancer [126], colon cancer [127], and gastric cancer [128]. However, histology analyses of these human tumors reveal that RHAMM expression is heterogeneous, and high RHAMM expression is often limited to a small subset of cancer cells [119]. The consequence of RHAMM to tumor progression may therefore depend upon both the cancer cell subtype that is expressing RHAMM and the spatial context of these RHAMM+ cells within the tumor (e.g., in invasive or metastatic niches). It is important to note that in some contexts, RHAMM is a tumor suppressor. For example, genomic reduction of RHAMM is associated with the initiation and aggression of malignant peripheral nerve sheath tumors [129]. When RHAMM is viewed as a multifunctional protein capable of interactions with a wide range of binding partners that can result in cell-specific functions, these opposing results are not necessarily surprising but highlight the critical importance of in-depth studies to provide a roadmap for targeting RHAMM in appropriate cancer types. Despite this apparent functional complexity, RHAMM is an attractive therapeutic target due to its restricted homeostatic expression predicting a good safety profile, its association with a poor outcome in most cancers, and its critical functions that promote inflammation and fibrosis [12,62], which exacerbate cancer progression [130]. Furthermore, and most importantly, experimental analyses described below using RHAMM peptide mimetics to disrupt cell surface HA-RHAMM signaling have unexpectedly revealed that this extracellular interaction may mastermind the multiple functions of RHAMM by providing the initiating and activating signal for its downstream functional complexity (Figure 8).

## 6. Therapeutic Strategies for Targeting RHAMM

In addition to its multifunctional nature, interest in using RHAMM as a therapeutic target in cancer has been reduced by the additional challenge of its largely coiled-coil secondary structure, which is generally difficult to target using traditional therapeutics such as small molecule inhibitors. Small molecule inhibitors have been successfully used for abrogating uncontrolled receptor activation and cell signaling in cancer. However, these function by binding, sometimes irreversibly, into the catalytic pockets of proteins to inhibit protein–ligand interactions and signal activation [131]. They are therefore not ideal reagents for binding to unstructured coiled-coil regions [131]. However, peptides that mimic the HA-binding region of RHAMM and compete with cell-surface RHAMM for HA have been shown to subsequently block the multiple functions of RHAMM in experimental models of disease [62,83]. For example, they blunt the signaling required for macrophage infiltration into injured tissues such as excisional wounds [104] and bleomycin-damaged lungs [62,132], thereby reducing tissue inflammation and destruction. These peptides also reduce both fibroblast motility and myofibroblast differentiation to consequently diminish fibrosis during excisional repair [104] and experimentally induced scleroderma [90]. Therefore, blocking this initial signal may be a reasonable approach for controlling the multifunctional complexity of aberrant RHAMM signaling during the progression of some cancers. Although RHAMM peptide mimetics have not yet been extensively tested for their ability to reduce tumorigenesis, they have been reported to reduce melanoma tumorigenesis [133] and breast and prostate cancer cell invasion [84,134] and proliferation in vitro and in vivo [135]. Since the complex processes of inflammation, fibrosis, and cancer invasion, proliferation, and metastasis require coordinated cell motility, microtubule/cytoskeleton dynamics, and gene expression, these results are consistent with a model whereby cell surface RHAMM/HA interactions activate and regulate multiple downstream intracellular and extracellular RHAMM functions. The successful use of RHAMM/HA binding peptide mimetics in controlling complex diseases in experimental models also argues for an effectiveness in blunting the most lethal aspect of some cancers, which is metastatic disease.

## 7. Conclusions

Targeting multifunctional proteins such as RHAMM that are only transiently expressed with tissue injury but chronically upregulated with disease are underutilized as therapeutic targets. However, relating their domain structure to function and defining domain functional hierarchy within specific tissue and disease contexts is critical for therapeutic success. This review has summarized the current knowledge on RHAMM domain structure and function and highlighted the critical importance of RHAMM/HA interactions as an activating signal for many multiple functions of this protein. Further research is required to determine if and when this activating signal promotes or suppresses progression of specific cancers.

## Figures and Tables

**Figure 1 ijms-22-10313-f001:**
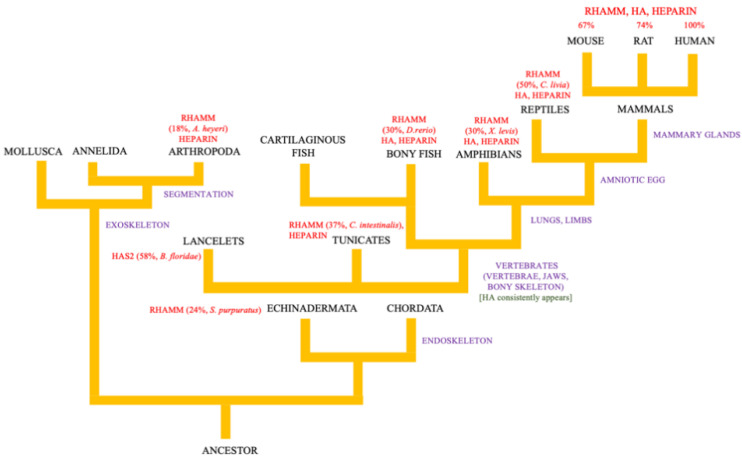
Evolution of RHAMM. The tree shows the appearance of hyaluronan binding ability of RHAMM with the appearance of vertebrates. More information about the evolutionary expression of RHAMM: https://www.ncbi.nlm.nih.gov/protein/?term=rhamm (accessed: 20 September 2021) and HAS2: https://www.ncbi.nlm.nih.gov/protein/?term=HAS2 (accessed: 20 September 2021) within various taxonomic groups.

**Figure 2 ijms-22-10313-f002:**
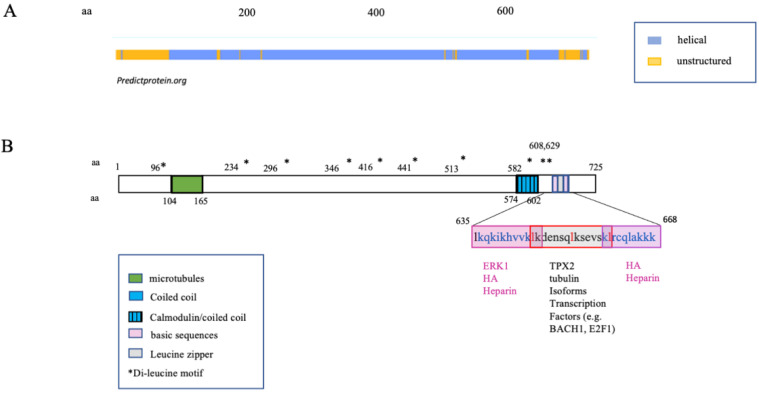
The predicted secondary structure of human RHAMM (**A**) and the known binding domains for which binding partners have been experimentally verified. (**A**) The secondary structure of RHAMM is predicted to be mostly helical (predictprotein.org (accessed: 18 September 2021)) and (**B**) the major binding domains identified to date occur in the alpha-helical carboxyl terminus. * *p* value < 0.05, ** *p* value < 0.01.

**Figure 3 ijms-22-10313-f003:**
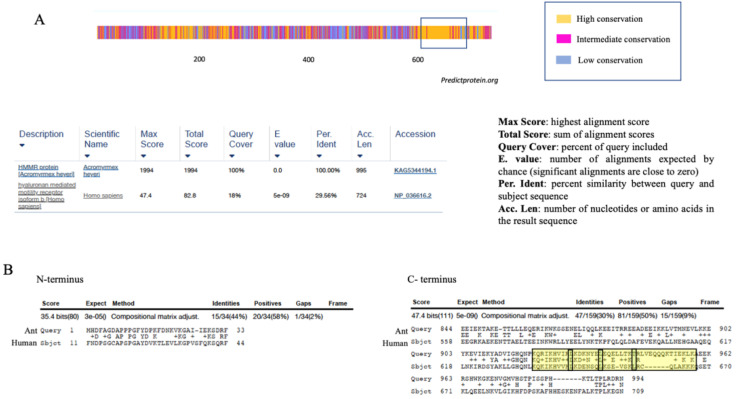
Homology of human RHAMM with invertebrates. (**A**) The conservation of RHAMM protein is predicted to be highest in the carboxyl-terminus (predictprotein.org (accessed: 18 September 2021)) that contains the basic amino acid clusters required for binding to HA and heparin (open box). (**B**) Sequence identity and homology (https://blast.ncbi.nlm.nih.gov/ (accessed: 18 September 2021)) with Acromymex heyeri, an invertebrate ant, is low and, as predicted in A, occurs primarily in the C-terminus containing basic amino acid clusters required for binding to hyaluronan and heparin in human RHAMM protein. The leucine zipper in this region (double-boxed) typical of vertebrate RHAMM and predicted to bind to transcription factors, is not present.

**Figure 4 ijms-22-10313-f004:**
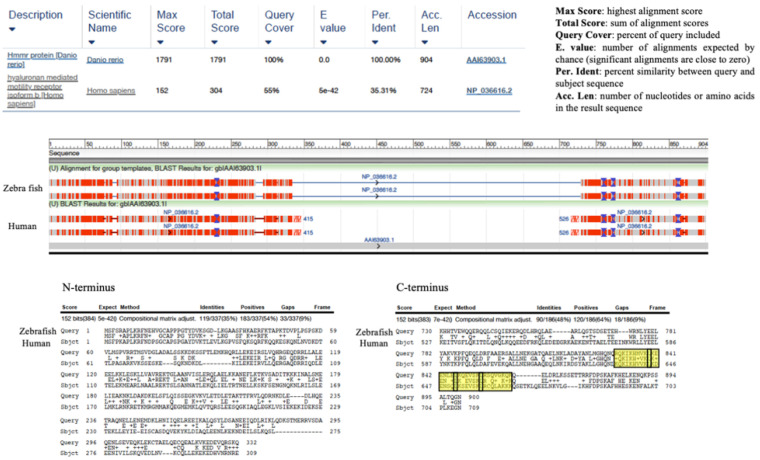
Sequence homology between human *RHAMM* and Danio rerio (zebrafish). The sequence homology in the N- and C-termini has increased relative to human *RHAMM* from that observed in invertebrates, particularly in the C-terminal hyaluronan/heparin-binding domain. The leucine zipper, which is predicted to bind to transcription factors, is present.

**Figure 5 ijms-22-10313-f005:**
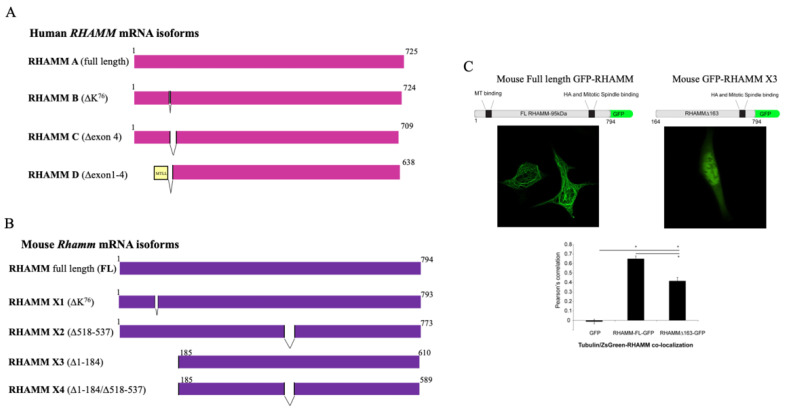
Isoform structures of RHAMM in (**A**) human and (**B**) mouse genomes. RHAMM isoforms are generated by alternative splicing and/or alternate start site/codon usage. (**C**) Full-length mouse GFP-RHAMM decorates interphase microtubules, while mouse GFP-RHAMM X3 is more diffuse and accumulates in the nucleus.

**Figure 6 ijms-22-10313-f006:**
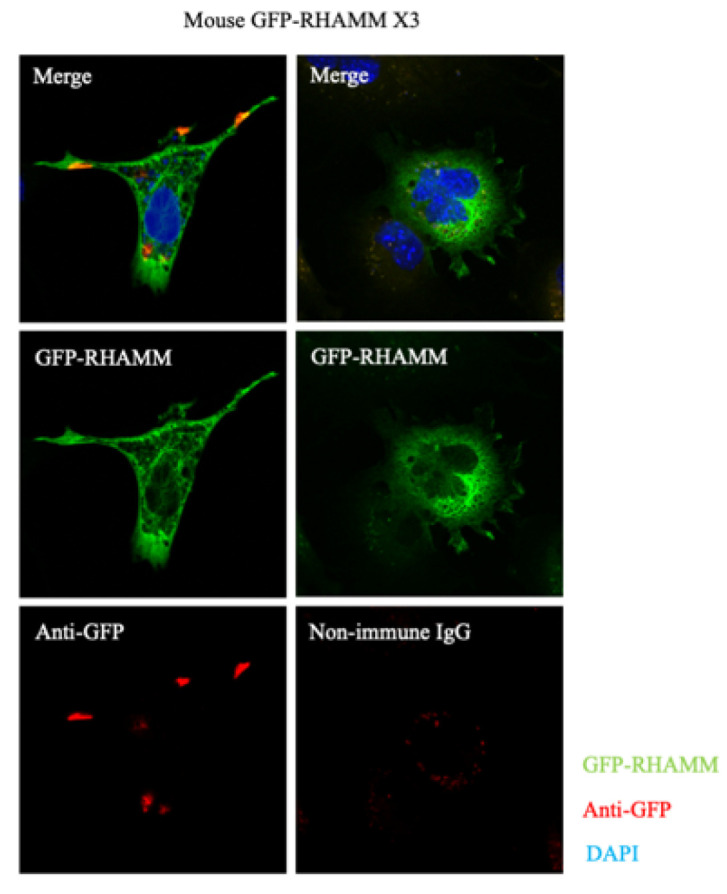
Subcellular localization of GFP-RHAMM expressed in mouse fibroblasts. Non-permeabilized cells exhibit diffuse localization for GFP-RHAMM X3. Anti-GFP antibodies were used to detect cell-surface RHAMM, which occurs in cell processes.

**Figure 7 ijms-22-10313-f007:**
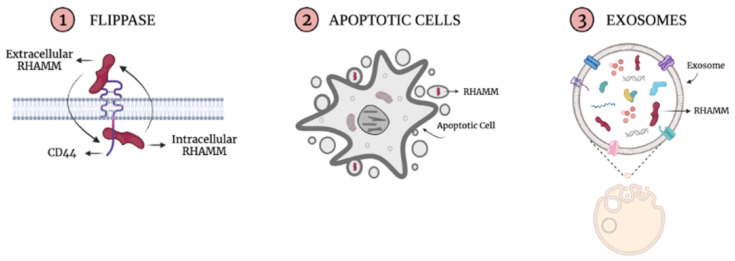
The appearance of RHAMM at the cell surface in the absence of an encoded signal peptide, transmembrane domain, or GPI-linker predicts it is exported unconventionally by using exosomes or flippase mechanisms and/or released by apoptotic cells. Once exported via unconventional mechanisms, RHAMM can interact with cell surface receptors, such as CD44, to facilitate its extracellular signaling functions.

**Figure 8 ijms-22-10313-f008:**
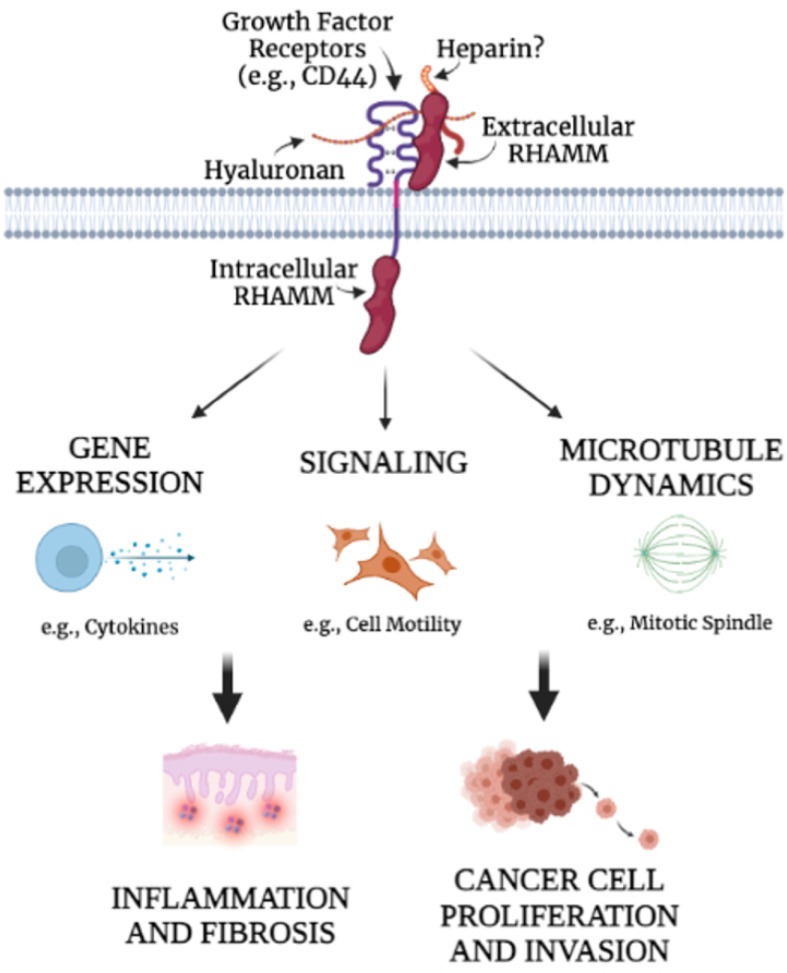
The interaction between cell surface RHAMM and HA is proposed to activate the multiple functions of extracellular and intracellular RHAMM. Since RHAMM plays a critical role in many HA-mediated functions, using reagents to target the HA-binding region of RHAMM is expected to reduce the pathologies associated with RHAMM activation.

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
