# Peer review of "RHAMM Is a Multifunctional Protein That Regulates Cancer Progression"

_ijms, 2021, doi:10.3390/ijms221910313_

Round 1

Reviewer 1 Report

This manuscript provides a well written and detailed description of the complex evolution of RHAMM and its extensive normal and pathological functions, including its interactions with hyaluronan.

Minor edits and some questions

(l. 39) ‘overlooked, yet it is emerging

(l. 48) ‘compartments, and its’

(l. 69) ‘in the sizes of the HA polymers’

(l. 75) ‘and b-glucuronic acid’

(l. 76) ‘inflammation, but it is’

(l. 86) ‘human synthases’

(l. 111) There is no ‘(ants, 18%)’ in figure 1. Also in lower right of figure 1, ‘SKELETON’

(l. 112) ‘(yellow Box, Figure 2).’ Note: the texts in figures 2 and 3 are very small and hard to read. The figures could be enlarged, and figure 4 moved to the next page where its legend is.

(l. 145) ‘with, and/or modifies the functions of’

(l. 162) ‘of basic amino acids (klk and klr) separated … zipper (Figure 5, yellow)’

(l. 187) ‘synthases, and RHAMM’

(l. 189-197) A model slide could help here.

(l. 202) ‘[92], but one’

(l. 252-253) ‘but expressions of … and isoforms are’

(l. 260) ‘genomes.’

(l. 262) Is (uniport.org) a site that identifies the mRNA splicing?

(l. 278) ‘C-termini are’

(l. 283) ‘confirm this’

(l. 326) ‘RHAMM’

(l. 329) ‘within the tumor’

(l. 337) ‘due to its’

Reviewer 2 Report

The review about RHAMM is focused on highlighting the multifunctionality of the protein rather than its role as hyaluronan receptor and this view is a novelty in the RHAMM papers. The manuscript although very interesting and reporting several novel information about RHAMM has some problems in the sequences of the paragraphs and the figures.

Here there is a list of suggestion for making the paper more flowing:

  • lines 78-83/figure 1: in the text figure 1 seems to refer to HA, but really is about RHAMM, can the authors add in the figure the appearance of HA in the tree?
  • - figure 1: there should be a link in the Chordata tree?
  • line 111: the reference to ants  is for figure 1 but is not clear, so I suggest to indicate it in the figure or change the text to "figure2"
  • line 112:in brackets there is "box figure 2" but this does not correspond to figure
  • line 122: can the authors add also other vertebrates (e.g. birds, bovine...) to confirm the statement?
  • lines 122-125: the statement is not connected to the cited figures 1 and 2; moreover, considerations about RHAMM domains here are not  easy to understand, they should be discussed after paragraph 3
  • figure 4: not well connected to the text in lines 122/3; in particular, part B can be moved with figure 6. The description report also names that are not explained before paragraph 4, so maybe they can be revised.
  • because of the complexity of the description of domains and functions, I suggest moving the 3rd paragraph after the 4th.
  • figure 5: it seems very scanty with respect to the text, for example, there are no indications of N- and C- terminus, positions of the different molecules binding sites in the scheme, or numbers of residues in the various modules shown
  • line 159: add the meaning of O-HA
  • line 189: transform figure 4 in figure 5
  • line 252: what does it mean "RHAMM expression is restricted in normal tissue"?
  • line 262: the bracket "uniport.org" is a reference?
  • there is no correspondence between figure 6 and the description in lines 263-265. Add indication of the various isoform (A-D) in the figure.
  • because of the description in lines 291-303, I suggest transforming fig 4B in 6B
